# Hepatotropic Properties of SARS-CoV-2—Preliminary Results of Cross-Sectional Observational Study from the First Wave COVID-19 Pandemic

**DOI:** 10.3390/jcm10040672

**Published:** 2021-02-09

**Authors:** Hanna Wiśniewska, Karolina Skonieczna-Żydecka, Miłosz Parczewski, Jolanta Niścigorska-Olsen, Ewa Karpińska, Monika Hornung, Krzysztof Jurczyk, Magdalena Witak-Jędra, Łukasz Laurans, Katarzyna Maciejewska, Łukasz Socha, Agnieszka Leonciuk, Dorota Bander, Malwina Karasińska-Cieślak, Bogusz Aksak-Wąs, Marta Wawrzynowicz-Syczewska

**Affiliations:** 1Department of Infectious Diseases, Hepatology and Liver Transplantation, Pomeranian Medical University, 71-455 Szczecin, Poland; hankaaa.wisniewska@gmail.com (H.W.); ewakarpinska@interia.pl (E.K.); monika.a.hornung@gmail.com (M.H.); jurczykkrzysztof@interia.pl (K.J.); asklepiada@wp.pl (Ł.L.); theville@wp.pl (Ł.S.); dbander@interia.pl (D.B.); 2Department of Biochemical Sciences, Pomeranian Medical University, 71-460 Szczecin, Poland; karzyd@pum.edu.pl; 3Department of Infectious, Tropical Diseases, and Immune Deficiency, Pomeranian Medical University, 71-455 Szczecin, Poland; milosz.parczewski@pum.edu.pl (M.P.); j.niscigorska-olsen@wp.pl (J.N.-O.); magdalenka86.mwj@gmail.com (M.W.-J.); kasiamaciejewsk@gmail.com (K.M.); celina.aga@gmail.com (A.L.); malwina_k_cieslak@tlen.pl (M.K.-C.); bogusz.aw@gmail.com (B.A.-W.)

**Keywords:** COVID-19, SARS-CoV-2 infection, liver enzyme abnormality, liver dysfunction

## Abstract

Liver injury—expressed as elevated liver enzymes—is common in patients with COVID-19. Little is known about the potential mechanisms of liver damage by SARS-CoV-2. A direct cytopathic effect on hepatocytes as well as injury related to hypoxia or hepatotoxicity are being considered. The aim of the study was to compare the clinical characteristic of COVID-19 disease in patients with normal and abnormal liver enzymes activity. A group of 150 patients with COVID-19, hospitalized in our center, was analyzed. Patients with the known liver comorbidities were excluded (*n* = 15). Clinical features and laboratory parameters were compared between patients with normal and abnormal aminotransferase values. Liver injury expressed as any alanine aminotransferase (ALT) elevation was noted in 45.6% of patients hospitalized due to COVID-19. The frequencies of aspartate aminotransferase (AST) elevation were lower. It was noted that elevated ALT/AST unfavorably affected other parameters related to liver function such as albumin level; gamma-glutamyl transpeptidase (GGTP); and partly, ALP activity and influenced inflammation-related parameters. The most probable cause of mild hepatitis during COVID-19 was anoxia and immune-mediated damage due to the inflammatory response following SARS-CoV-2 infection. A direct cytopathic effect of SARS-CoV-2 on hepatocytes, albeit less probable, can be considered as well. The use of potentially hepatotoxic drugs may contribute to liver damage.

## 1. Introduction

In late December 2019, China reported a cluster of severe pulmonary infections of unknown cause in Wuhan City, Hubei Province. A novel β-coronavirus officially named SARS-CoV-2 (severe acute respiratory syndrome coronavirus 2; previously: novel coronavirus or 2019-nCoV) emerging in humans was confirmed as the cause of this severe acute respiratory syndrome, subsequently named coronavirus disease 2019 (COVID-19, previously: 2019-nCoV acute respiratory disease) [1,2,3,4,5]. Researchers consider horseshoe bats as the most likely natural reservoir for SARS-CoV-2. SARS-CoV-2 has been spreading rapidly throughout the world—on 11 March 2020, the World Health Organization (WHO) declared the COVID-19 outbreak a global pandemic [5]. At present, the SARS-CoV-2 pandemics remains uncontrolled. On 3 November 2020, the total number of COVID-19 cases was 45,968,799, with 1,192,911 deaths (mortality rate 2.56%) [6]. In Poland, there were 362,731 confirmed cases, with 5631 deaths (mortality rate 1.55%). The true mortality rate of COVID-19 is still unknown [6]. Forecasting and modeling suggest that infection numbers will contribute to a rise globally in the forthcoming months [7]. Two major transmission routes were established: person-to-person (direct contact, most often via small droplets produced by coughing, sneezing, and talking) or indirect transmission (noncontact, by contaminated surfaces and objects). The incubation period ranges from 1 to 14 days, usually from 3 to 7 days, with the median of 5.5 days [4]. The virus causing the current pandemic was identified to be genetically related to SARS-CoV (severe acute respiratory syndrome coronavirus) and MERS-CoV (Middle East respiratory syndrome-related coronavirus), with a greater epidemic potential, higher infectivity, and less prominence, while all—SARS-CoV, MERS-CoV, and SARS-CoV-2—have been associated with severe, acute respiratory symptoms in humans [1,2,4,7]. COVID-19 is characterized by rapid epidemic transmissions due to a lack of herd immunity and notable mortality, increasing with age and among patients with comorbidities [7]. The symptoms of COVID-19 are nonspecific, ranging from asymptomatic cases, through mild (as a self-limiting respiratory disease) to extremely severe, progressive pneumonia, which can be fatal [3,4,5]. Most people infected with COVID-19 present mild symptoms [6]. Typical clinical manifestations of COVID-19 are the following: fever, dry cough, shortness of breath, fatigue and muscle pain, no improvement on antibiotic treatment, loss of sense of taste or smell, diarrhea, low white blood cell count such as neutropenia and lymphopenia, and pneumonia [3,4]. Pulmonary symptoms dominate in the clinical presentation of COVID-19 as the lungs are the main target of SARS-CoV-2, but it may also involve other organs, causing various organ dysfunctions [4,8]. SARS-CoV-2 utilizes the angiotensin-converting enzyme 2 (ACE-2) receptor for the cellular entry [8,9]. ACE-2 protein is widely expressed on the surface of multiple types of human cells—including alveolar epithelial cells in the lungs; bronchial epithelial cells; nasal and oral mucosa; as well as nasopharynx, cardiovascular system, guts (ACE-2 is abundantly present in the enterocytes of all parts of the small intestine, including the duodenum, jejunum, and ileum but not enterocytes of the colon), kidneys, central nervous system, adipose tissue, and the liver [4,8,10,11,12]. The ACE-2 receptors in the liver are mainly expressed on cholangiocytes in human ductal organoids, minimally expressed on hepatocytes and absent on Kupffer cells [8,13]. Liver injury was previously reported in 14–53% of SARS-CoV-2-infected patients [14]. The mechanism of liver damage is poorly recognized, and several possibilities need to be considered. It is uncertain whether the COVID-19-related liver dysfunction is due to the viral infection of liver cells or is secondary to coexisting conditions such as the use of potentially hepatotoxic drugs, systemic inflammatory response, disseminated intravascular coagulation (DIC), respiratory distress syndrome-induced hypoxia, and multiple organ dysfunction [15]. The liver appears to be one of the most frequently affected organ by SARS-CoV-2 [12,16,17,18]. The aim of our study was to compare clinical characteristic of COVID-19 disease in patients with normal and abnormal liver enzymes activity to evaluate the possible mechanisms of liver injury, especially the relation with oxygen dependence, inflammatory parameters, and drugs used for antiviral treatment. We hypothesized that liver damage in COVID-19 might be due to inflammation.

## 2. Experimental Section

### 2.1. Materials and Methods

For this retrospective, single-center, cross-sectional observational study, we recruited patients hospitalized from 1 March to 28 May 2020 in the Marie Curie Regional Hospital of Szczecin, Poland, a reference COVID-19 hospital for our region. The Pomeranian Medical University Ethics committee positively evaluated the study (KB-0012/85/12/2020/Z). Sensitive data were pseudonymized and encrypted in order to ensure identity protection. All obtained research data in the process were identified on the basis of the numerical code of the test object. Access to personal data was only granted to the minimum number of people necessary to conduct a scientific research (medical doctors serving as pandemic front-line medical workers) with the appropriate authorizations to process personal data.

One hundred and fifty consecutive patients that were admitted to our department were included in the study. Nasopharyngeal swab specimens were obtained from all patients at admission. In all cases, COVID-19 was confirmed by a real-time reverse transcription polymerase chain reaction (real-time RT-PCR) assay for SARS-CoV-2. Diagnosis and treatment of COVID-19 were guided according to the protocol issued by the Polish Association of Epidemiologists and Infectologists [3]. Co-infections with other respiratory viruses such as influenza A virus, influenza B virus, and respiratory syncytial virus were excluded. To evaluate the possible mechanisms of liver injury during SARS CoV-2 infection, cases with concomitant liver disease were excluded from further analyses (Figure 1).

The medical records of patients were analyzed by the research team of the Department of Infectious Diseases, Hepatology, and Liver Transplantation, Pomeranian Medical University, Szczecin, Poland; by the Department of Infectious, Tropical Diseases, and Immune Deficiency, Pomeranian Medical University, Szczecin, Poland; and by the Department of Human Nutrition and Metabolomics, Pomeranian Medical University, Szczecin, Poland. Recorded information included medical history, epidemiological data, underlying comorbidities, signs and symptoms on admission, laboratory test and chest computed tomography (CT) results, treatment measures including treatment history, and clinical outcomes during hospital stay. The analyzed clinical outcomes included history of oxygen use, the necessity for mechanical ventilation, as well as discharge and survival data. Laboratory confirmation of SARS-CoV-2 and the other laboratory tests as well as imaging diagnostics were all performed locally.

Liver damage was assessed on the basis of alanine aminotransferase (ALT) and aspartate aminotransferase (AST) activity. For the purpose of our analysis, three ALT/AST measurements were taken into account: baseline values, checked on the day of hospital admission, the highest level of ALT/AST during hospitalization, and the level of aminotransferases on the day of discharge after completion of treatment. The upper limit of ALT and AST normal was 32 U/L for women and 41 U/L for men. In our analyses, we categorized patients as “normal” and “abnormal” regarding AST and ALT values at admission and during hospital stay.

For cholestatic pattern of liver injury, alkaline phosphatase (ALP; range of normal 35–105 U/L) and gamma-glutamyl transpeptidase (GGTP, range of normal 6–41 U/L) were analyzed. Liver synthetic function was assessed using serum albumin concentration (range of normal 3.5–5.2 g/dL), total serum bilirubin (range of normal 0–1.2 mg/dL), and international normalization ratio (INR, range of normal 0.8–1.2) on admission. An albumin level below the lower limit of normal; elevated bilirubin, GGTP, and ALP above the upper limit of normal; and prolonged INR were considered parameters related to liver function. Inflammation and nonspecific damage from hypoxia were assessed using parameters such as C-reactive protein (CRP, normal value < 5.0 mg/L), D-dimers (range of normal 0–500 FEU ug/L), ferritin (range of normal 13–150 ng/mL), interleukin 6 (IL6, normal value < 7.0 pg/mL), lactate dehydrogenase (LDH, range of normal 135–214 U/L), lymphocyte count (range of normal 0.6–3.4 G/L), and white blood cell count (WBC, range of normal 4.00–10.00 G/L). The dynamics of aminotransferase changes in relation to cholestatic parameters, synthetic function parameters, and parameters of inflammation and nonspecific damage from hypoxia were evaluated.

We also divided the study patients based on body mass index (BMI) into those with normal weight (BMI 18.5–24.99 kg/m^2^), overweight (BMI 25.00–29.99 kg/m^2^), obesity class I (BMI 30–34.99 kg/m^2^), obesity class II (BMI 35.00–39.99 kg/m^2^), and obesity class III (BMI over 40.00 kg/m^2^) [19].

Oxygen saturation below 94% was considered an indication for oxygen supplementation (usually at flow of 4 L/min at least 4 times a day for 10 min and, additionally, if necessary). Data on oxygen dependence were collected. As far as oxygen dependence is considered, the patients were divided into two groups: persons not requiring oxygen therapy and those requiring oxygen supply (simple oxygen supplementation and invasive ventilation).

We analyzed the dynamics of AST and ALT alterations during the study period, that is, from admission until discharge along with their normality regarding different parameters of inflammation and liver damage, as described. Additionally, AST/ALT alterations during hospitalization in relation to oxygen dependence were evaluated. The treatment regimen and oxygen dependence were also compared.

### 2.2. Statistical Analyses

The distribution of continuous variables was evaluated by the Shapiro–Wilk normality test. The variables were presented as numbers (percentages) or medians with interquartile ranges (IQR). Consequently, nonparametric tests were used, i.e., the Kruskal–Wallis and Mann–Whitney tests, as appropriate. For studying the dynamics of aminotransferase concentration (repeated measures), the Friedman test was applied. For the correlation analyses, the Spearman rank method was used. By means of logistic regression, we modeled the probabilities AST and ALT abnormality. A full output is given in Appendix A. The two-tailed *p* < 0.05 was adopted as statistically significant. Analyses were performed in MedCalc^®^ Statistical Software version 19.6 (MedCalc Software Ltd., Ostend, Belgium; https://www.medcalc.org (accessed on 8 November 2020); 2020). The false discovery rate (FRD) method was used to control type I errors. The calculations were performed using the *p*-adjusted function of the stats package in R studio (https://cran.r-project.org (accessed on 8 November 2020)) [20]. A priori sample size analyses was not but effect sizes and post hoc power analyses were conducted in the G*Power software [21,22].

## 3. Results

### 3.1. Study Subjects

A total of 150 study persons were enrolled, predominantly females (*n* = 82, 54.7%), with the median age of 55 (IQR: 42.0–62.0) years. The youngest study participant was 19, whilst the oldest was 87 years old. In 46 patients (30.7%), no coexisting diseases were reported. Fifteen patients were excluded from further analyses due to concomitant liver disease, among them, *n* = 8 (5.3%) had nonalcoholic fatty liver disease (NAFLD), *n* = 5 (3.3%) had alcoholic liver disease (ALD), *n* = 1 (0.7%) had acute hepatitis A infection, and *n* = 1 (0.7%) had liver and kidney dysfunction in the course of systemic lupus. The most frequent concomitant diseases were hypertension (45 patients; 33.33%) and heart diseases (21 patients; 15.56%) including ischemic heart disease, atrial fibrillation, and valve regurgitation. In 15/135 (11.36%) cases, glucose intolerance and, in 12/135 (8.89%), hypothyroidism were present. Twelve out of 135 patients (8.89%) had a history of malignant tumors (breast cancer—*n* = 3; thyroid cancer—*n* = 3, colorectal cancer—*n* = 2, prostate cancer—*n* = 1, pancreatic cancer—*n* = 1, uterine cancer—*n* = 1, and mycosis fungoides—*n* = 1). Five (5/135; 3.7%) patients suffered from respiratory disease, most commonly bronchial asthma. As a result, a dataset of 135 patients (including 75 females, 55.6%) with a median age of 55 (IQR: 41.25–62) years was analyzed. Among the analyzed patients, 63 (46.67%) had normal weight, 41 (30.37%) were overweight, 26 (19.26%) had obesity class I, 3 (2.22%) had obesity class II, and 2 (1.48%) had obesity class III. No patient in the study group (*n* = 135) had been previously diagnosed with nonalcoholic steatohepatitis (NASH)/nonalcoholic fatty liver disease (NAFLD). Patients were hospitalized for a median of 13 (IQR: 9.9–16.7) days. The majority of patients reported respiratory symptoms (*n* = 96; 71.1%), e.g., cough, dyspnea, and pain in the chest. There were no significant differences for the prevalence of respiratory or gastric symptoms across genders (*p* > 0.05). Mortality in the group of patients without liver disease was 2.22% (3/135 patients). Missing data were ignored. The numbers of persons with particular parameters are reported within the Results section.

### 3.2. Pharmacological Treatment by Oxygen Use

A total of 33 (24.5%) patients required oxygen supplementation; among them, a group of 29 patients (21.5%) received simple oxygen supplementation (for instance and 4 (3.0%) were subjected to the invasive ventilation. Lopinavir/ritonavir was administered to the majority of patients on oxygen therapy (Table 1). Among the patients who required oxygen (both invasive ventilation and simple oxygen supplementation), AST activity was significantly higher compared to persons with no oxygen demand; however, after multiple comparisons, no significant differences were found. The results are shown in Table 2. The patients were divided into a group that did not require oxygen and a group of patients who, at some point during hospitalization, required oxygen supply (oxygen supply we understand as any kind of oxygen therapy).

### 3.3. The Dynamics of AST and ALT Activity

AST and ALT values varied significantly over time. ALT concentration was significantly higher during the treatment compared to baseline and endpoint values. AST was significantly lower at endpoint when compared to baseline and during the hospital stay. We found that these values were affected by gender, with females presenting significantly lower enzyme activities in all but two cases, i.e., AST at discharge (*p* = 0.06) and ALT during treatment (*p* = 0.09). The results are shown in Table 3 and in Figure 2 and Figure 3.

### 3.4. The Dynamics of AST and ALT Activity

In the present study, we assumed that the upper limits of ALT and AST normal were 32 U/L for women and 41 U/L for men. At admission, there were 37/135 patients (27.41%) with abnormal AST (median value 46 and maximum 116 U/L), with 22/75 females (29.33%) and 15/60 males (25%) present abnormal AST. Regarding ALT at admission, there were 44/135 (32.59%) patients with elevated ALT (median value 47 and maximum 133 U/l), with 23/75 females (30.67%) and 21/60 males (35%) present elevated ALT. During hospitalization, the median for abnormal AST was 48 (maximum 268) U/L and the median for abnormal ALT was 60 (maximum 397) U/L. In total, 35/135 (25.93%) patients, including 21/75 females (28%) and 14/60 males (23.33%), presented abnormal AST and 61/135 (45.61%) patients, including 32/75 females (42.67%) and 29/60 males (48.33%), showed abnormal ALT values. At discharge, there were 27/135 (20%) patients with abnormal AST, including 18/75 females (24%) and 8/60 males (15%), and 50/135 (37.04%) subjects with elevated ALT, including 25/75 females (33.33%) and 25/60 males (41.67%). We found that the frequencies of aminotransferase abnormalities were not influenced by gender (*p* > 0.05). Elevated AST on admission and during treatment was unfavorably associated with albumin and GGTP concentrations (Table 4). Additionally, when AST was abnormal at admission, there were statistical tendencies toward higher ALP (*p* = 0.08) and INR (*p* = 0.07). For inflammation-related parameters, we discovered that CRP, D-dimers, ferritin, IL-6, LDH, and WBC were significantly higher in patients with elevated AST at admission and during treatment, with the latter only significant for the first aminotransferase evaluation prior to hospitalization.

The results were partly replicated regarding ALT abnormalities, with no significant linkage to albumin at both study points. ALP concentrations were significantly higher in persons with ALT abnormal levels at admission only. CRP, ferritin, and LDH were significantly higher when ALT was abnormal both at admission and during treatment. For IL-6, we found a significantly higher concentration in the case of ALT abnormalities during treatment only. The results are shown in Table 4 and Table 5.

In the logistic regression model (backward method), we found that LDH activity predicted abnormality within AST at admission ((b = 0.015, SE = 0.05, Wald = 7.32, *p* = 0.06, and Exp (b) = 1.01, with 95% CI = 1.0043 to 1.0269) and that ferritin predicted AST abnormality during treatment (b = 0.0047, SE = 0.002, Wald = 7.67, *p* = 0.005, and Exp (b) = 1.0047, with 95% CI = 1.00 to 1.008).

### 3.5. Correlation between AST and ALT at Baseline, during the Treatment, and at Hospitalization Endpoint with Biochemical Parameters

A negative correlation between the albumin concentration and AST activity at admission, during Tx, and at discharge was found, while for GGTP, CKMB, CRP, D-dimers, ferritin, IL-6, and LDH, the correlations were positive. For ALT values, the GGTP, CRP, D-dimers, ferritin, IL-6, and LDH correlations assessed at all three study points were positive, with D-dimers and WBC associations with ALT significant only during treatment, and at admission and during treatment, respectively (Table 6).

### 3.6. Correlation between ALT/AST Activity and Pharmacological Treatment

We observed a significantly higher ALT/AST activity in patients taking lopinavir/ritonavir during treatment and at discharge, as shown in Table 7.

## 4. Discussion

This report, to our knowledge, is the largest case series to date of hospitalized patients with COVID-19 in Poland.

In our series, approximately one third of patients infected with SARS-CoV-2 presented with liver injury, expressed as ALT and, less frequently, AST elevation at admission. This elevation was usually mild and asymptomatic. ALT at admission did not exceed 132 U/L, and the highest value of AST was 116 U/L. Median ALT/AST values were higher in males in comparison with females; however, frequencies of ALT/AST abnormalities were similar in both genders. In none of the patients, mild hepatitis was accompanied by the impairment of liver function; however, AST, but not ALT, tended to be negatively correlated with albumin levels at admission and during Tx. Low albumin and elevation of liver enzymes may indicate a more severe course of SARS-CoV-2 infection [23]. During hospitalization, the number of patients with ALT/AST abnormalities slightly increased, but again, there was no case of liver decompensation. At discharge, the frequency of patients with abnormal aminotransferases decreased and a tendency for normalization was more evident for AST than for ALT. It may be partly explained by the influence of potentially hepatotoxic drugs on ALT value.

The mechanism of liver injury during COVID-19 is not fully understood. According to recent findings, approximately 40% of US patients, infected with SARS-CoV-2, present with some sort of hepatitis [24]. This means that the liver may be one of the most frequently occupied organs by the virus [25]. It was shown that liver enzymes elevation in the course of COVID-19 is associated with disease severity [26]. The expression of ACE receptors on hepatocytes and abundant expression on some cholangiocytes suggests that liver injury could be mediated via bile duct cells [27]. It is worth noting that elevated aminotransferases at admission and during hospitalization did not affect but were associated with markers of cholestasis in our series. Very few liver biopsies, performed in COVID-19 patients, do not allow for an unambiguous description of hepatitis pattern. Specimens taken from deceased patients showed moderate microvesicular steatosis and mild lobular and portal activity, indicating that liver injury can be partly related to the hepatotoxicity of drugs used for COVID-19 treatment and partly to direct SARS-CoV-2 injury (viral hepatitis) [28]. This needs further clarification. On the other hand, we found a significant correlation between skewed aminotransferases and inflammation-related parameters such as CRP, D-dimers, ferritin, IL-6, LDH, and WBC at admission and during hospital stay. It was more evident for AST than for ALT. Moreover, significantly higher AST but not ALT was found in the patients with oxygen dependence and a more severe clinical course of COVID-19. The latter suggests that liver injury is secondary to the inflammatory response and hypoxia rather than to viral injury, and AST contribution from sources outside the liver, primarily from muscles, has to be also considered.

Another reason for aminotransferase elevation is a drug-induced liver injury. The lack of a standard-of-care in COVID-19 forced us to experiment with a set of drugs such as antiviral agents (lopinavir/ritonavir and chloroquine), antibiotics (azithromycin and cephalosporins), and biological drugs (tocilizumab). We managed to show that ALT activity was significantly higher in patients treated with chloroquine and lopinavir/ritonavir and tended to be higher in patients on azithromycin. In patients hospitalized with confirmed SARS-CoV-2 infection, treatment with chloroquine and lopinavir-ritonavir did not yield appreciable benefits. In new recommendations, chloroquine and lopinavir/ritonavir are not recommended. It was also found that, in oxygen-dependent patients, lopinavir/ritonavir was used more frequently than in patients not requiring oxygen supply and that it was linked to higher AST activity. It can be interpreted as hypoxic hepatitis in the first instance, but hepatotoxicity of these antiviral agents cannot be ruled out. A higher frequency of patients with abnormal ALT activity at discharge than at admission indicates that drug-induced liver injury is a probable explanation.

## 5. Conclusions

The most probable cause of mild hepatitis during COVID-19 in our series was anoxia and immune-mediated damage due to inflammatory response following SARS-CoV-2 infection. A direct cytopathic effect of SARS-CoV-2 on hepatocytes, albeit less probable, needs to be taken into account as well. The use of potentially hepatotoxic drugs may contribute to liver damage. However, due to insufficient power in some analyses, more studies are needed to confirm our findings.

Our study, however, has some limitations. The most important weakness of this study and, generally, many studies on liver injury in the course of COVID-19 is the lack of histopathological examinations that might confirm or rule out direct hepatic cytopathy of SARS-CoV-2. Therefore, our conclusions on the possible mechanisms of COVID-19 hepatitis are speculative and confounded by many factors such as the potential hepatotoxicity of drugs used for treatment, the influence of hypoxia on liver function, and hepatic reaction on systemic inflammation.

Overall, our study shows that the liver is an organ frequently affected by SARS-CoV-2 during COVID-19 infection; thus, liver enzymes as well as liver function parameters should be carefully evaluated in hospitalized patients and monitored thereafter.

## Figures and Tables

**Figure 1 jcm-10-00672-f001:**
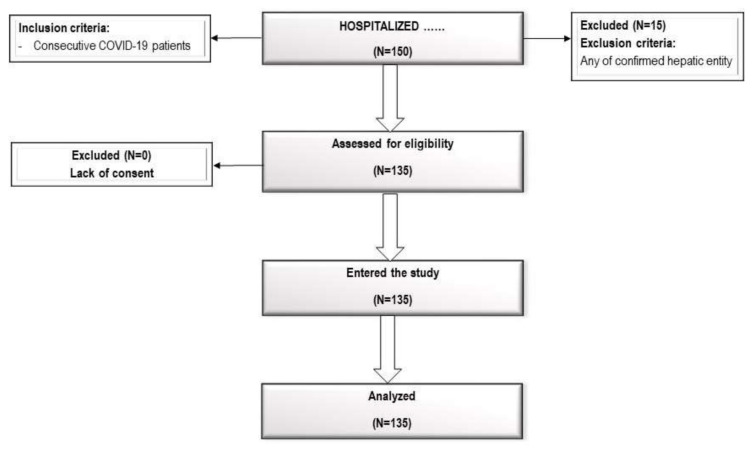
Study flow chart.

**Figure 2 jcm-10-00672-f002:**
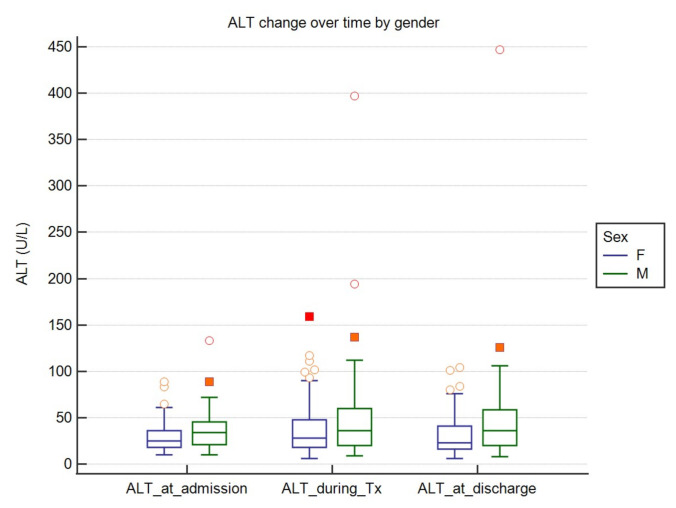
ALT (alanine aminotransferase) change over time by gender. Middle lines represent medians and central boxes stand for IQRs. Tx—treatment. Dots and squares represent *outside* and *far out* values.

**Figure 3 jcm-10-00672-f003:**
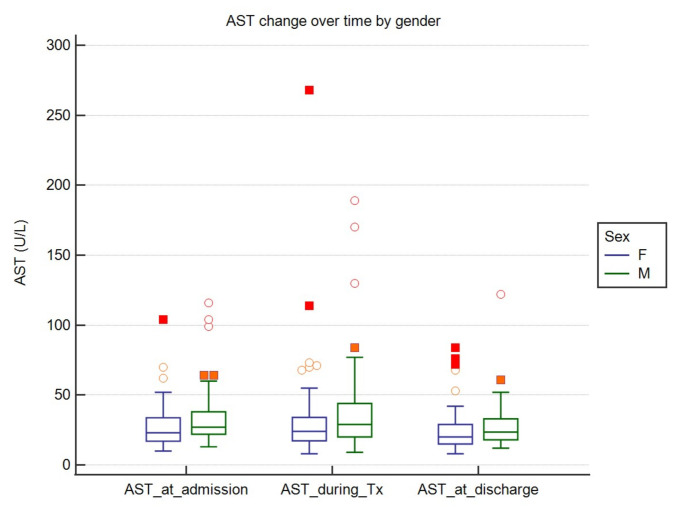
AST (aspartate aminotransferase) change over time by gender. Middle lines represent medians and central boxes stand for IQRs. Tx-treatment. Dots and squares represent *outside* and *far out* values.

**Table 1 jcm-10-00672-t001:** Pharmacological treatment by oxygen dependence (patients without oxygen supply, simple oxygen supplementation, and invasive ventilation).

Drug	Patients without Oxygen Supply *n* = 102	Simple Oxygen Supplementation *n* = 29	Invasive Ventilation *n* = 4	P *(FDR)
**Azithromycin (Y/N)**	98/4 (96.08%/3.92%)	29/0 (100%/0%)	4/0 (100%/0%)	0.51
**Chloroquine (Y/N)**	89/13 (87.25%/12.75%)	29/0 (100%/0%)	4/0 (100%/0%)	0.15
**Lopinavir/Ritonavir (Y/N)**	5/97 (4.9%/95.1%)	12/17 (41.38%/58.62%)	2/2 (50%/50%)	0.00003
**Tocilizumab (Y/N)**	0/102 (0%/100%)	0/29 (0%/100%)	1/3 (25%/75%)	n.a.

* regarding all oxygen-dependent groups. Y—yes, N—no, n.a.—not applicable.

**Table 2 jcm-10-00672-t002:** Liver enzyme activity relative to oxygen dependence (a group of patients requiring oxygen supply at any time during hospitalization).

Variable	O_2_ Therapy Patient Group Requiring Oxygen Supply (*n* = 33)			No O_2_ Therapy Required Oxygen-Free Patient Group (*n* = 102)			P (FDR)
	Median	IQR	95%CI for median	Median	IQR	95%CI for median	
ALT at admission (U/L)	25	20–38	21.411–35.589	26.5	20–40	22.0–35.0	0.9449
ALT during Tx (U/L)	38	20–66	24.0–58.356	30	19–50	26.0–37.0	0.3438
ALT at discharge (U/L)	28	23–49	24.0–39.589	25	16–47	23.0–31.123	0.3438
AST at admission (U/L)	25	22–36	22.0–28.0	25	18–35	18.411–30.0	0.3438
AST during Tx (U/L)	30	22–49	21.0–27.374	23	18–34	24.411–39.178	0.135
AST at discharge (U/L)	23	18–38	19.0–24.0	22	16–29	18.411–30.0	0.3438

Legend: ALT—alanine aminotransferase; AST—aspartate aminotransferase; patients were divided into a group that did not require oxygen and a group of patients who, at some point during hospitalization, required an oxygen supply (oxygen supply we understand as any kind of oxygen therapy); FDR—false detection rate, IQR—interquartile range, CI—confidence interval.

**Table 3 jcm-10-00672-t003:** The dynamics of transaminase concentration during the hospital stay.

Variable	All Patients (*n* = 135)			Women (*n* = 75)			Men (*n* = 60)			P (FDR)
	Median	IQR	Maximum	Median	IQR	95% CI for Median	Median	IQR	95% CI for Median	
ALT at admission (U/L)	26	20–39.75	133	25.0	18.0–36.0	21.540–27.0	34.0	21.0–45.5	23.0–39.0	0.0306
ALT during treatment (U/L)	33	19–56	397	28.0	18.0–47.750	23.080–37.0	36.0	20.0–60.0	30.0–49.0	0.0954
ALT at discharge (U/L)	25	17.25–47	447	23.0	16.250–41.0	20.0–26.0	36.0	20.0–58.5	25.0–42.243	0.0306
AST at admission (U/L)	25	19–35	116	23.0	17.0–33.750	19.540–27.460	27.0	22.0–38.0	24.0–31.061	0.05
AST during treatment (U/L)	25	19–38.75	268	24.0	17.250–34.0	20.0–27.0	29.0	20.0–44.0	23.0–32.061	0.05
AST at discharge (U/L)	22	16–30	122	20.0	15.0–29.0	17.540–22.0	23.5	18.0–33.0	20.939–27.0	0.0596

Legend: ALT—alanine aminotransferase; AST—aspartate aminotransferase; FDR—false detection rate, IQR—interquartile range, CI—confidence interval.

**Table 4 jcm-10-00672-t004:** Biochemical parameters in relations to AST normality at admission and during the treatment.

Variable	Abnormal AST at Admission				Normal AST at Admission				P	Power (d)	Abnormal AST during Tx				Normal AST during Tx				P (FDR)	Power (d)
	N	Median	IQR	95% CI for Median	n	Median	IQR	95% CI for Median			N	Median	IQR	95% CI for Median	n	Median	IQR	95% CI for Median		
Liver function related parameters																				
**Albumin (g/L)**	12	3.35	2.800–3.650	2.473–3.683	22	4.05	3.700–4.200	3.891–4.200	0.014	0.9 (1.1)	16	3.45	3.250–3.700	3.264–3.700	18	4.15	3.900–4.200	3.940–4.200	0.00325	0.9 (1.1)
**GGTP (U/L)**	33	64	42.750–93.250	47.233–77.123	88	24	16.0–43.0	22.0–30.668	<0.01	0.93 (0.66)	37	64	32.250–93.250	46.387–79.459	84	24	16.500–43.0	22.0–31.456	0.03	0.94 (0.65)
**CRP (mg/L)**	36	19.845	7.295–91.650	9.256–67.890	99	8.2	2.855–18.512	4.284–10.632	0.002	0.85 (0.54)	39	20.4	9.060–92.055	11.456–81.380	96	6.605	2.600–16.795	3.948–9.576	0	0.97 (0.71)
**D-dimers (FEU ug/L)**	29	558	372.0–1091.500	433.042–901.086	76	391	320.500–638.500	360.867–461.033	0.025	0.53 (0.38)	32	580	408.500–1146.500	454.993–921.125	73	386	322.750–629.250	352.654–460.383	0.01244	0.55 (0.38)
**Ferritin (ng/mL)**	24	453	150.0–1066.0	232.140–665.100	54	162	87.100–330.0	107.991–241.706	0.002	0.56 (0.45)	24	490.5	246.0–1263.500	336.166–991.982	54	150	87.100–309.0	107.991–229.367	0.03	0.6 (0.48)
**IL6 (pg/mL)**	28	17.45	6.600–39.250	7.651–34.625	56	8.3	3.450–17.0	5.820–11.002	0.001	0.36 (0.29)	31	19.7	7.425–51.550	13.903–38.865	53	6.8	3.375–13.125	5.318–8.820	0.08	0.39 (0.32)
**LDH (U/L)**	36	235	201.0–352.500	209.318–298.070	97	185	161.500–221.250	175.372–198.628	0.025	0.99 (0.94)	39	222	198.500–353.250	207.917–305.323	94	185	160.0–222.0	175.023–198.977	0.03	0.99 (0.9)
**WBC (G/L)**	36	6.47	5.505–7.720	5.900–7.493	97	5.23	4.137–7.050	4.740–5.951	0.025	0.58 (0.37)	39	5.96	4.117–7.677	5.256–7.114	94	5.615	4.270–7.140	5.061–6.350	0.6748	0.12 (0.09)

Legend: ALT—alanine aminotransferase; AST—aspartate aminotransferase; ALP—alkaline phosphatase; GGTP—gamma-glutamyl transpeptidase; INR—international normalization ratio, CRP—C-reactive protein; IL6—interleukin 6; LDH—lactate dehydrogenase; WBC—white blood cell; d—Cohen’s effect size; IQR—interquartile range, CI—confidence interval.

**Table 5 jcm-10-00672-t005:** Biochemical parameters in relations to ALT normality at admission and during the treatment.

Variable	Abnormal ALT at Admission (U/L)				Normal ALT at Admission (U/L)				P (FDR)	Power (d)	Abnormal ALT during Tx (U/L)				Normal ALT during Tx (U/L)				P (FDR)	Power (d)
	n	Median	IQR	95% CI for Median	n	Median	IQR	95% CI for Median			N	Median	IQR	95% CI for Median	n	Median	IQR	95% CI for Median		
Liver Function Related Parameters																				
**GGTP (U/L)**	39	61	36.500–82.50	45.752–72.083	82	22.5	15.0–43.0	20.652–28.0	<0.01	0.82 (0.52)	56	51	28.500–80.0	36.185–65.0	65	23	15.0–40.250	21.0–28.872	0.06	
**Parameters of Inflammation and Nonspecific Damage from Hypoxia**																				
**CRP (mg/L)**	44	12.07	4.725–67.645	8.206–20.362	91	8.87	2.957–25.567	4.752–12.914	0.157	0.41 (0.27)	39	20.4	4.302–76.370	8.896–20.494	96	6.605	2.430–18.720	4.256–9.722	0.06	0.85 (0.48)
**Ferritin (ng/mL)**	26	398	150.0–560.0	248.785–535.866	52	162	87.400–303.0	108.397–239.948	0.022	0.12 (0.12)	36	382.5	127.0–551.500	161.202–508.705	42	153.5	92.700–279.0	107.555–243.596	0.0276	0.65 (0.15)
**IL6 (pg/mL)**	28	13.8	5.450–24.400	6.430–22.162	56	8.35	3.700–20.0	6.099–12.804	0.446	0.27 (0.24)	45	14.4	6.375–26.700	8.192–19.562	39	6.8	2.775–15.0	4.075–11.233	0.0884	0.24 (0.22)
**LDH (U/L)**	44	211	176.0–285.500	198.069–224.965	89	192	161.500–230.0	176.780–206.220	0.056	0.86 (0.52)	61	212	182.250–292.0	198.752–225.0	72	184.5	156.0–222.0	174.212–199.788	0.0017	0.98 (0.67)

Legend: ALT—alanine aminotransferase; AST—aspartate aminotransferase; ALP—alkaline phosphatase; GGTP—gamma-glutamyl transpeptidase; INR—international normalization ratio’ CRP—C-reactive protein; IL6—interleukin 6; LDH—lactate dehydrogenase; WBC—white blood cell; FDR—false detection rate, IQR—interquartile range, CI—confidence interval.

**Table 6 jcm-10-00672-t006:** Correlation between AST and ALT at baseline, during the treatment, and at hospitalization endpoint with biochemical parameters.

Variable		ALT at Discharge (U/L)	ALT during Tx (U/L)	ALT at Admission (U/L)	AST at Discharge (U/L)	AST during Tx (U/L)	AST at Admission (U/L)
**Liver Function Related Parameters**							
**Total Bilirubin (mg/dl) (*n* = 108)**	**Correlation coefficient**	−0.138	−0.085	0.011	0.004	0.002	0.07
	**Significance Level P**	0.1556	0.3793	0.9137	0.9692	0.9849	0.4689
	**P (FDR)**	0.933	0.9378	0.9849	0.9849	0.9849	0.9378
**GGTP (U/L) (*n* = 121)**	**Correlation coefficient**	0.496	0.513	0.61	0.509	0.569	0.643
	**Significance Level P**	<0.0001	<0.0001	<0.0001	<0.0001	<0.0001	<0.0001
	**P (FDR)**	0.00001	0.00001	0.00001	0.00001	0.00001	0.00001
**Parameters of Inflammation and Nonspecific Damage from Hypoxia**							
**CRP (mg/L) (*n* = 135)**	**Correlation coefficient**	0.282	0.318	0.192	0.286	0.44	0.381
	**Significance Level P**	0.0009	0.0002	0.0257	0.0008	<0.0001	<0.0001
	**P (FDR)**	0.0010	0.00004	0.0257	0.0011	0.00003	0.00003
**D-dimers(FEU ug/L) (*n* = 105)**	**Correlation coefficient**	0.109	0.194	0.036	0.198	0.377	0.316
	**Significance Level P**	0.2694	0.0471	0.712	0.043	0.0001	0.001
	**P (FDR)**	0.3232	0.0706	0.712	.0706	0.00006	0.003
**Ferritin (ng/mL) (*n* = 78)**	**Correlation coefficient**	0.38	0.456	0.465	0.454	0.574	0.586
	**Significance Level P**	0.0006	<0.0001	<0.0001	<0.0001	<0.0001	<0.0001
	**P (FDR)**	0.00006	0.0001	0.0001	0.0001	0.0001	0.0001
**IL6 (pg/mL) (*n* = 84)**	**Correlation coefficient**	0.282	0.278	0.2	0.398	0.419	0.414
	**Significance Level P**	0.0093	0.0105	0.0687	0.0002	0.0001	0.0001
	**P (FDR)**	0.0126	0.0126	0.0687	0.00004	0.00003	0.00003
**LDH (U/L) (*n* = 133)**	**Correlation coefficient**	0.358	0.446	0.371	0.382	0.556	0.548
	**Significance Level P**	<0.0001	<0.0001	<0.0001	<0.0001	<0.0001	<0.0001
	**P (FDR)**	0.00001	0.00001	0.00001	0.00001	0.00001	0.00001
**Lymphocytes (G/L) (*n* = 135)**	**Correlation coefficient**	−0.047	−0.08	0.107	−0.109	−0.146	−0.069
	**Significance Level P**	0.5887	0.3555	0.2159	0.2061	0.0904	0.4238
	**P (FDR)**	0.5887	0.5085	0.4318	0.4318	0.4318	0.5085
**WBC (G/L) (*n* = 133)**	**Correlation coefficient**	0.089	0.163	0.236	0.011	0.084	0.166
	**Significance Level P**	0.3062	0.0602	0.0063	0.8975	0.3344	0.056
	**P (FDR)**	0.4013	0.1204	0.0378	0.8975	0.4013	0.1204

Legend: ALT—alanine aminotransferase; AST—aspartate aminotransferase; ALP—alkaline phosphatase; GGTP—gamma-glutamyl transpeptidase; INR—international normalization ratio, CRP—C-reactive protein; IL6—interleukin 6; LDH—lactate dehydrogenase; WBC—white blood cell; FDR—false detection rate, IQR—interquartile range, CI—confidence interval.

**Table 7 jcm-10-00672-t007:** Correlation between ALT/AST activity and pharmacological treatment.

**Variable**	**Chloroquine YES**			**Chloroquine NO**			**P (FDR)**
	**N**	**Median**	**IQR**	**N**	**Median**	**IQR**	
ALT_at_discharge	122	26.0000	19.000–49.000	13	17.0000	11.000–41.000	0.179
ALT_during_Tx		34.5000	20.000–60.000		19.0000	13.750–46.250	0.1372
AST_at_discharge		22.0000	16.000–34.000		18.0000	15.000–28.250	0.284
AST_during_Tx		25.5000	19.000–39.000		20.0000	16.500–29.000	0.1372
**Variable**	**Azithromycin YES**			**Azithromycin NO**			**P (FDR)**
	**N**	**Median**	**IQR**	**n**	**Median**	**IQR**	
ALT_at_discharge	131	26.0000	18.250–47.750	4	14.0000	11.500–28.500	0.1804
ALT_during_Tx		34.0000	19.250–57.500		14.5000	13.000–31.000	0.1804
AST_at_discharge		22.0000	16.000–30.750		18.0000	15.500–23.500	0.1804
AST_during_Tx		25.0000	19.000–39.000		18.0000	14.500–26.000	0.1804
**Variable**	**Lopinavir/Ritonavir YES**			**Lopinavir/Ritonavir NO**			**P (FDR)**
	**N**	**Median**	**IQR**	**N**	**Median**	**IQR**	
ALT_at_discharge	19	40.0000	27.750–87.250	116	24.0000	16.500–45.000	0.0036
ALT_during_Tx		63.0000	31.500–90.000		30.0000	19.000–49.000	0.0046
AST_at_discharge		30.0000	19.000–40.750		21.5000	16.000–29.000	0.0092
AST_during_Tx		38.0000	28.500–69.750		23.5000	18.500–33.500	0.0036

FDR—false detection rate, IQR—interquartile range.

## Data Availability

The data presented in this study are available on request from the corresponding author. The data are not publicly available due to their containing information that could compromise the privacy of research participants.

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
