# Peer review of "Hepatotropic Properties of SARS-CoV-2—Preliminary Results of Cross-Sectional Observational Study from the First Wave COVID-19 Pandemic"

_jcm, 2021, doi:10.3390/jcm10040672_

Round 1

Reviewer 1 Report

This clinical retrospective study attempts to compare the fate of COVID-19 patients with elevated or normal liver enzyme. The research hypothesis is interesting. The manuscript is well written, except some typo and missing references. However, there is some dramatic limitations and concerns in the design of this study.

First, there is no information about ethics. Did the authors obtain the authorization of any ethic committee for this work? What about data protection? Please ad a statement for this.

Moreover, the authors are presenting their work as a retrospective study, but some laboratory value are clearly not used in routine such as IL-6.

Secondly, the primary outcome is not clear and I saw no sample size calculation. The recruitment period seems to be arbitrary. Why choosing one month, the authors could have taken in account all admitted patients?

Except liver disease what were the other inclusion/exclusion criteria? Moreover, there is no data about the BMI of patients. Indeed, the prevalence of NAFLD is so high in patient with metabolic syndrome that we should expect that very high BMI category could be considered as a surrogate of NAFLD. Did you exclude or include these patients?

Finally, I’m not familiar with the FDR approach but why did you not start with a multivariate analysis with logistic regression? This is a standard approach that should be done if the application conditions are respected and I’m sure they are. (If they are not you can ad a supplementary material for this).

In details:

p.2 l.50-51: references are missing

p.2 78-79 and p.12 l.287: what about expression of ACE on liver sinusoidal endothelial cells and on gallbladder epithelium? Some reports demonstrate the presence of the virus in gallbladder wall (PMID: 33238410, PMID: 32890595). Liver sinusoid endothelial cells are an important source of IL-6 in liver injury following platelet recruitment (PMID: 32443494), in several reports autopsy unveiled sinusoid dilatation and thrombi (PMID: 32792598, PMID: 32654359. You could develop this in the discussion part.

P.7 l. 204 When you are reporting the different proportions, please, mention the confidence interval; especially if these are compared.

p.4 l.177 correct the typo

Table 3 it is not clear what do you compare with the FDR statistical test

l.245 I suggest to not use abbreviation in a subtitle

l.283-284 ref?

Author Response

Reviewer 1

We wish to thank the Editor and Reviewer for the thoughtful and detailed comments concerning our manuscript. We have responded to each concern below

This clinical retrospective study attempts to compare the fate of COVID-19 patients with elevated or normal liver enzyme. The research hypothesis is interesting. The manuscript is well written, except some typo and missing references. However, there is some dramatic limitations and concerns in the design of this study.

First, there is no information about ethics. Did the authors obtain the authorization of any ethic committee for this work? What about data protection? Please ad a statement for this.

The Pomeranian Medical University Ethics committee has positively evaluated the study (no. of approval was added within the body of manuscript – KB-0012/85/12/2020/Z). Sensitive data was pseudonymised and encrypted in order to ensure their protection. All obtained research data in the process of their processing was identified on the basis of the numerical code of the test object. Access to personal data was only granted to the minimum number of people necessary to conduct a scientific research (medical doctors serving as pandemic front line medical workers), having the appropriate authorizations to process personal data.

Moreover, the authors are presenting their work as a retrospective study, but some laboratory value are clearly not used in routine such as IL-6.

We have included IL6 evaluation, as this parameter is routinely used in the hospital settings in Poland.

Secondly, the primary outcome is not clear and I saw no sample size calculation. The recruitment period seems to be arbitrary. Why choosing one month, the authors could have taken in account all admitted patients?

We have amended the primary goal of the study for clarity. For the study purpose we have selected first 150 hospitalized patients, however identity number was randomly given. As to a few study aims regarding hepatotropic properties of SARS CoV 2, we did not perform sample size calculation to ensure proper statistical power. However, we have done such analyses post-hoc and added the information within the text. 

Except liver disease what were the other inclusion/exclusion criteria? Moreover, there is no data about the BMI of patients. Indeed, the prevalence of NAFLD is so high in patient with metabolic syndrome that we should expect that very high BMI category could be considered as a surrogate of NAFLD. Did you exclude or include these patients?

Regarding BMI, we have included such data within materials and methods section.

On admission to the ward, each patient had a CT scan of the chest. Everyone's liver was assessed - some patients were radiologically diagnosed with hepatic steatosis. Patients with hepatic steatosis were excluded from the study. We also divided our patients into obese and non-obese patients. Some of the obese patients had hepatic steatosis (as I wrote above, patients with hepatic steatosis were excluded from the study). 

Finally, I’m not familiar with the FDR approach but why did you not start with a multivariate analysis with logistic regression? This is a standard approach that should be done if the application conditions are respected and I’m sure they are. (If they are not you can ad a supplementary material for this).

We have added logistic regression as requested. FDR was approached for multiple comparisons, meaning regarding all variables in particular analyses.

In details:

p.2 l.50-51: references are missing

 This was amended within the text.

p.2 78-79 and p.12 l.287: what about expression of ACE on liver sinusoidal endothelial cells and on gallbladder epithelium? Some reports demonstrate the presence of the virus in gallbladder wall (PMID: 33238410, PMID: 32890595). Liver sinusoid endothelial cells are an important source of IL-6 in liver injury following platelet recruitment (PMID: 32443494), in several reports autopsy unveiled sinusoid dilatation and thrombi (PMID: 32792598, PMID: 32654359. You could develop this in the discussion part.

Thank you for a such valuable comment. We have added the appropriate section within the text.

P.7 l. 204 When you are reporting the different proportions, please, mention the confidence interval; especially if these are compared.

Thank you. We have amended that where appropriate.

p.4 l.177 correct the typo

We have amended that.

Table 3 it is not clear what do you compare with the FDR statistical test

 FDR was approached for multiple comparisons, meaning regarding all variables in particular analyses.

l.245 I suggest to not use abbreviation in a subtitle

We have corrected that.

l.283-284 ref?

We have revised that section accordingly.

Reviewer 2 Report

Many thanks for your hard work on producing this work. I think there is some useful information here but that the reader almost drowns in the tidal wave of data. It could benefit from some reduction. Furthermore I recommend following the STROBE guidelines. My specific points are:

Introduction

  1. Lines 47-69 of the introduction are completely superfluous and can be discarded (from "Researchers consider... ....and lymphopenia, pneumonia (3,4)"
  2. Line 72 - please replace ref 8 with "Cell, Vol. 181, Issue 2, p271–280.e8" for a stronger reference on SARS-CoV-2 viral entry, please
  3. Line 84 - the liver doesn't truly appear to be the second organ affected after the lungs as there is prominent digestive symptomatology in the majority of cases (gut), and kidney injury is also very common in hospitalised SARS-CoV-2 patients. Moderate the comment please
  4. In line with STROBE guidelines (please ensure these are followed throughout the manuscript) please state a hypothesis that was being tested

Methods

  1. Line 123 - CKMB cannot really be described as a non-specific marker of "damage from hypoxia" as it is unclear what the proximal cause of increase in this cardiomyocyte cell death marker is. Please alter the terminology for this marker.
  2. Lines 131-132 "However to evaluate... ...analyses." would be much better placed at the end of line 98 so that the first paragraph of the methods essentially informs the reader who is included and excluded fro the study.

Results

  1. Lines 162-165 with the excluded patients should be presented before the breakdown of comorbidities. Comorbidities should then be presented with 135 patients as the denominator, not 150. Again, in keeping with STROBE recommendations a flowsheet should be considered for use to show study flow.
  2. At the end of section 3.1 mortality data needs to be presented for this cohort of 135
  3. Lines 174 & 175 - I believe the terminology needs to be changed to non-invasive ventilation and invasive ventilation. Is this correct or have I misunderstood? invasive oxygen flow and non-invasive oxygen therapy are ambiguous
  4. What do the authors mean by the abbreviation f.i. in parentheses on line 174?
  5. Do the authors means in table 1 for headings to be non-invasive ventilation and invasive ventilation? If so, please correct as non-invasive & invasive supply are ambiguous
  6. Table 2 dichotomises patients according to need for O2 - is this need for O2 at admission or at any time in the stay? Please clarify and add the explanation to the legend and text
  7. Figures 1 & 2 should be provided as supplements. It is also questionable that the authors split patients up according to gender as they fail to comment on the significance of this in the discussion. My impression is that the authors should not dichotomise according to gender as this will also simplify the section of results from lines 208-218
  8. Line 219 - change "unfavourably affected..." to "was unfavourably associated with..."
  9. For Table 6 it is appropriate to exclude parameters which showed no significant differences in Tables 4 & 5 (i.e. ALP, Br, INR) and this will streamline the data which is already very heavy in the paper
  10. I think the section on pharmacological treatment needs more work. The only treatment that makes sense to analyse is Lopinavir/Ritonivir (L/R) because for the other drugs almost 100% of every severity group received the drugs. However the findings of transaminase abnormalities in the L/R group should be further analysed within the O2 only groups as there was a near 50/50 split of who received the drugs there i.e. are the LFT differences in the main analysis of L/R confounded by disease severity?

Discussion

  1. Lines 282-4 need a reference
  2. Lines 284-5 needs a reference ("it was shown that....")
  3. Lines 285-7. The abundant expression of ACE2 only occurs in a small percentage of cholangiocytes (see J Hep 73(4), P993-995, 2020). The following reference doesn't need including but you may find it relevant: A Single-Cell RNA Expression Map of Human Coronavirus Entry Factors. Cell Reports. 2020. 32(12), 108175.
  4. Line 288 - again it must be stated that the aminotransferases did not affect, but were associated with markers of cholestasis
  5. Lines 303-4 - I understand you only showed significance for lopinavir/ritonavir, not chloroquine.
  6. Lines 310-2 "Higher frequency......indicates that drug-induced liver injury is a probable explanation."What about residual/ongoing damage which is in resolution? Surely it's not possible to pinpoint a cause?
  7. Please consider the strengths and weaknesses of your paper and what new information it adds to the literature (other than it being the largest hospital dataset from Poland)

Conclusion

Please address your conclusion in light of the hypothesis you will draw up at the outset of the paper

Author Response

Reviewer 2

We wish to thank the Editor and Reviewer for the thoughtful and detailed comments concerning our manuscript. We have responded to each concern below

  1. Lines 47-69 of the introduction are completely superfluous and can be discarded (from "Researchers consider... ....and lymphopenia, pneumonia (3,4)"

The text was amended as suggested.

  1. Line 72 - please replace ref 8 with "Cell, Vol. 181, Issue 2, p271–280.e8" for a stronger reference on SARS-CoV-2 viral entry, please.

We have changed the citation.

  1. Line 84 - the liver doesn't truly appear to be the second organ affected after the lungs as there is prominent digestive symptomatology in the majority of cases (gut), and kidney injury is also very common in hospitalised SARS-CoV-2 patients. Moderate the comment please

This was our mental shortcut, we meant the digestive tract as a whole (including the liver). We improved the content for clarity.

  1. In line with STROBE guidelines (please ensure these are followed throughout the manuscript) please state a hypothesis that was being tested

We added a hypothesis and evaluated the text as to STROBE guidelines.

Methods

  1. Line 123 - CKMB cannot really be described as a non-specific marker of "damage from hypoxia" as it is unclear what the proximal cause of increase in this cardiomyocyte cell death marker is. Please alter the terminology for this marker.

This was changed as suggested. Thank you.

  1. Lines 131-132 "However to evaluate... ...analyses." would be much better placed at the end of line 98 so that the first paragraph of the methods essentially informs the reader who is included and excluded fro the study.

We have amended this part accordingly.

Results

  1. Lines 162-165 with the excluded patients should be presented before the breakdown of comorbidities. Comorbidities should then be presented with 135 patients as the denominator, not 150. Again, in keeping with STROBE recommendations a flowsheet should be considered for use to show study flow.

We have prepared a flow chart.

  1. At the end of section 3.1 mortality data needs to be presented for this cohort of 135

We have added this information.

  1. Lines 174 & 175 - I believe the terminology needs to be changed to non-invasive ventilation and invasive ventilation. Is this correct or have I misunderstood? invasive oxygen flow and non-invasive oxygen therapy are ambiguous

We have amended that.

  1. What do the authors mean by the abbreviation f.i. in parentheses on line 174?

We meant for instance.

  1. Do the authors means in table 1 for headings to be non-invasive ventilation and invasive ventilation? If so, please correct as non-invasive & invasive supply are ambiguous

We have changed that.

  1. Table 2 dichotomises patients according to need for O2 - is this need for O2 at admission or at any time in the stay? Please clarify and add the explanation to the legend and text

We have added appropriate explanations.

  1. Figures 1 & 2 should be provided as supplements. It is also questionable that the authors split patients up according to gender as they fail to comment on the significance of this in the discussion. My impression is that the authors should not dichotomise according to gender as this will also simplify the section of results from lines 208-218

We decided to split patients by gender, as there are major differences in liver-linked parameters regarding gender.

  1. Line 219 - change "unfavourably affected..." to "was unfavourably associated with..."

We have changed that.

  1. For Table 6 it is appropriate to exclude parameters which showed no significant differences in Tables 4 & 5 (i.e. ALP, Br, INR) and this will streamline the data which is already very heavy in the paper.

As suggested, we have removed insignificant differences data.

  1. I think the section on pharmacological treatment needs more work. The only treatment that makes sense to analyse is Lopinavir/Ritonivir (L/R) because for the other drugs almost 100% of every severity group received the drugs. However the findings of transaminase abnormalities in the L/R group should be further analysed within the O2 only groups as there was a near 50/50 split of who received the drugs there i.e. are the LFT differences in the main analysis of L/R confounded by disease severity?

Because there is  a lack of sufficient analyses on drugs impact on hepatic malfunctions in the current study, that will be further studied. We intend to assess the impact of the drugs used to treat COVID19 in the next publication. The publication will compare the treatments used to treat patients in the first and the second waves of COVID19.

Discussion

  1. Lines 282-4 need a reference

We have added a citation.

  1. Lines 284-5 needs a reference ("it was shown that....")

We have added a citation.

  1. Lines 285-7. The abundant expression of ACE2 only occurs in a small percentage of cholangiocytes (see J Hep 73(4), P993-995, 2020). The following reference doesn't need including but you may find it relevant: A Single-Cell RNA Expression Map of Human Coronavirus Entry Factors. Cell Reports. 2020. 32(12), 108175.

We have added this citation.

  1. Line 288 - again it must be stated that the aminotransferases did not affect, but were associated with markers of cholestasis

We have amended that.

  1. Lines 303-4 - I understand you only showed significance for lopinavir/ritonavir, not chloroquine.

Yes, correct, only for lopinavir/rotonavir.

  1. Lines 310-2 "Higher frequency......indicates that drug-induced liver injury is a probable explanation."What about residual/ongoing damage which is in resolution? Surely it's not possible to pinpoint a cause?

We have added appropriate extension on this paragraph.

  1. Please consider the strengths and weaknesses of your paper and what new information it adds to the literature (other than it being the largest hospital dataset from Poland)

We have added this section.

Conclusion

Please address your conclusion in light of the hypothesis you will draw up at the outset of the paper

We have addressed the conclusions in relation to hypothesis.

Round 2

Reviewer 1 Report

The authors addressed almost all the comments but some point remained unclear and deserve clarifications:

“We have included IL6 evaluation, as this parameter is routinely used in the hospital settings in Poland."

As this is not performed as a standard in other countries, could you please mention how this was measured (ELISA kit?) and the reference of the kit/device.

 “We have added logistic regression as requested. FDR was approached for multiple comparisons, meaning regarding all variables in particular analyses.”

Please ad the full output in supplementary material section

“Regarding BMI, we have included such data within materials and methods section. On admission to the ward, each patient had a CT scan of the chest. Everyone's liver was assessed - some patients were radiologically diagnosed with hepatic steatosis. Patients with hepatic steatosis were excluded from the study. We also divided our patients into obese and non-obese patients. Some of the obese patients had hepatic steatosis (as I wrote above, patients with hepatic steatosis were excluded from the study). “

I cannot find the mention of BMI in this section and I don’t understand what this variable has to do in this section as your exclusion criteria was steatosis and not BMI.

Moreover, chest CT-scan performed for such conditions (covid infection) usually don’t have a native protocol (whiteout iodine contrast) and you cannot rule out steatosis when iodinated contrast is injected. If you don’t’ want to use the BMI as a continuous variable, you can categorize it in known categories (overweight, obesity stade I, stade II, stade III etc.)

Moreover, the corrected sections have typos and language mistakes.

E.g.:

l.85 word missing: « organ »

l.102: something missing in the corrected sentence

l.163-164: please rewrite the sentence which is not grammatically correct.

186-188: sentence unclear. You can just state that missing data were ignored.

Author Response

Reviewer 1

Once again, we wish to thank the Editor and Reviewer for the thoughtful and detailed comments concerning our manuscript. We have responded to each concern below.

- “We have included IL6 evaluation, as this parameter is routinely used in the hospital settings in Poland." As this is not performed as a standard in other countries, could you please mention how this was measured (ELISA kit?) and the reference of the kit/device.

IL6 was assessed by immunochemiluminescence on a Cobas801 device. 

 - “We have added logistic regression as requested. FDR was approached for multiple comparisons, meaning regarding all variables in particular analyses.” Please ad the full output in supplementary material section

Full output data is attached. 

- “Regarding BMI, we have included such data within materials and methods section. On admission to the ward, each patient had a CT scan of the chest. Everyone's liver was assessed - some patients were radiologically diagnosed with hepatic steatosis. Patients with hepatic steatosis were excluded from the study. We also divided our patients into obese and non-obese patients. Some of the obese patients had hepatic steatosis (as I wrote above, patients with hepatic steatosis were excluded from the study). “ I cannot find the mention of BMI in this section and I don’t understand what this variable has to do in this section as your exclusion criteria was steatosis and not BMI. Moreover, chest CT-scan performed for such conditions (covid infection) usually don’t have a native protocol (whiteout iodine contrast) and you cannot rule out steatosis when iodinated contrast is injected. If you don’t’ want to use the BMI as a continuous variable, you can categorize it in known categories (overweight, obesity stade I, stade II, stade III etc.)

We have completed the article with information on body mass index (BMI).

We divided the patients based on body mass index (BMI) into those with normal weight BMI 18.5 - 24.99 kg/m2, overweight BMI 25.00 - 29.99 kg/m2, obese class I BMI 30 - 34.99 kg/m2, obese class II BMI 35.00 - 39.99 kg/m2 and obese class III BMI over 40.00 kg/m2.

There were no underweight  among the study participants.

In the group of 150 patients: 66 patients (44%) had normal body weight, 45 patients (30%) were overweight, 39 subjects were obese - 31 subjects (20.67%) had  obese class I, 5 subjects (3.33%) had obese class II, 3 subjects (2%) had obese class III.

We excluded 15 subjects with liver disease from further analysis. Among the excluded 15 subjects: 3 people had normal range of weight, 4 people were overweight, 8 people had obesity - in 5 people with obesity class I, 2 people with obesity class II, 1 person with obesity class III. Among those with liver disease, 8 had a history of steatosis or newly diagnosed steatosis on chest CT scan. All subjects with hepatic steatosis were found to be excessive body weight - 2 subjects were overweight, 4 subjects had obesity class I, 1 subject had obesity class II, and 1 subject had obesity class III.

After ruling out liver disease, a group of 135 subjects was further analysed. Among these individuals: 63 subjects (46.67%) had normal range of weight, 41 subjects (30.37%) were overweight, 26 subjects (19.26%) had obesity class I, 3 subjects (2.22%) had obesity class II, 2 subjects (1.48%) had obesity class III.

Despite excessive body weight in more than half of the study participants (excessive body weight 53.33% vs. normal weight 46.67%), no patient in the study group (n=135) had previously been diagnosed with non-alcoholic staetohepatitis (NASH) / non-alcoholic fatty liver disease (NAFLD). 8 from 15 patients excluded from the study had hepatic steatosis diagnosed before hospitalization.

 - E.g.:

l.85 word missing: « organ » We have changed that.

l.102: something missing in the corrected sentence - We have changed that.

l.163-164: please rewrite the sentence which is not grammatically correct.

To control type I errors false discovery rate (FDR) approach was used.

We have changed that.

186-188: sentence unclear. You can just state that missing data were ignored. We have changed that.

Reviewer 2 Report

I thank the authors for their improved manuscript. I only mention now that there is an amended "free hanging sentence" near the beginning of one of the paragraphs that makes no sense.

More importantly I still have some difficulty (may be just requires clarification) understanding the pulmonary illness severity of this cohort. The overwhelming majority do not seem to require oxygen which is in keeping with the low mortality rate. Nevertheless I wonder now why they were admitted to hospital - was it policy to admit all positive cases to hospital in Poland, or were these patients admitted for different reasons but coincidentally infected with the virus? Please specify.

Secondly, according to the table headings no patients received oxygen alone as any oxygen requirement resulted in either mechanical or non-invasive ventilation. This seems highly irregular and needs clarification and explanation - was this also policy at the hospital? If I have misunderstood and the non-invasive group actually were only receiving supplemental oxygen, then this requires relabelling, as non-invasive ventilation is not the same as the receipt of simple oxygen supplementation. Otherwise I have no further comments

Once these matters

Author Response

Reviewer 2

Once again, we wish to thank the Editor and Reviewer for the thoughtful and detailed comments concerning our manuscript. We have responded to each concern below.

- I thank the authors for their improved manuscript. I only mention now that there is an amended "free hanging sentence" near the beginning of one of the paragraphs that makes no sense.

We have moved the sentence to another place - as a continuation of the text.

- More importantly I still have some difficulty (may be just requires clarification) understanding the pulmonary illness severity of this cohort. The overwhelming majority do not seem to require oxygen which is in keeping with the low mortality rate. Nevertheless I wonder now why they were admitted to hospital - was it policy to admit all positive cases to hospital in Poland, or were these patients admitted for different reasons but coincidentally infected with the virus? Please specify.

At the beginning of the covid19 pandemic (the first wave of COVID19 outbreak in Poland), we did not know what the course of the disease would be. All patients with a positive smear result for SARS-CoV-2 were admitted to hospital. Most patients were in good general condition and mildly ill with COVID19 - this meant that only a small percentage of our patients required oxygen therapy. Patients requiring oxygen therapy were few in number. If a patient required oxygen therapy he or she was in a moderate to severe condition. The epidemic situation in Poland was under control - a small number of cases of illness and deaths were reported. There were only few cases of critical COVID19 in our hospital. We are currently working on publication of the second wave of COVID19 outbreak in Poland.  We thaught, that It will be a completely different study, because currently Polish hospitals do not hospitalize patients with a mild course of COVID19. At the moment, most of our hospitalized patients require oxygen therapy - invasife ventilation using high oxygen flows.

- Secondly, according to the table headings no patients received oxygen alone as any oxygen requirement resulted in either mechanical or non-invasive ventilation. This seems highly irregular and needs clarification and explanation - was this also policy at the hospital? If I have misunderstood and the non-invasive group actually were only receiving supplemental oxygen, then this requires relabelling, as non-invasive ventilation is not the same as the receipt of simple oxygen supplementation. Otherwise I have no further comments

Thank you very much for this valuable comment!

We labeled the tables unfortunate.

Our patients mostly did not require oxygen therapy (102 patients).

Among our subjects, 33 required any oxygen supply. 29 patients required simple oxygen supplementation. The 4 patients we describe in the article required treatment with invasive oxygen therapy. 

During the first wave of COVID19 (March-May 2020) we were just building experience, there were few cases of severe SARS-CoV-2 infection. Patients with a severe course went directly to the intensive care unit - they required ventilator therapy. At that time, in our hospital, simple oxygen supplementation was adequate and sufficient treatment for most patients. Currently, the number of intensive care places is limited, which has prompted us to introduce high-flow oxygen therapy more frequently.
